# OpenReview forum: "AgentDAM: Privacy Leakage Evaluation for Autonomous Web Agents"
_NeurIPS.cc/2025/Datasets_and_Benchmarks_Track — NeurIPS 2025 Datasets and Benchmarks Track poster_

### Official Review · Reviewer_pff7 · 2025-06-26

**Rating:** 5
**Confidence:** 3

**Summary:**

The authors address the timely problem of privacy leaks in Agentic LLms. Llms are increasingly being connected with private user data (in context, e.g., from files, history etc) and are increasingly able to perform actions on user's behalf (e.g., post online). This opens to the risk of leaking private data to external parties. The authors study a methodology to evaluate and detect the risk. The produce LLM-based models to perform actions using sensitive information and to evaluate if the actions performed where in line with reasonable expectations of the user in terms of data minimization. They perform analysis with various llms and in various domains. They release the code and data.

**Dataset Code Accessibility:**

Yes

**Ethical Considerations:**

No, there are no or only very minor ethics concerns

**Final Justification:**

I have reviewed carefully the rebuttal and other reviews and I confirm the assessment of the paper. the paper presents datasets for an important problem with low coverage of datasets and benchmarks and has sound methodology.

**Limitations Weaknesses:**

- difficulty in assessing the precision of the evaluation methodology. The authors could report more details on how precise is the LLM as a judge in evaluating the leakage.

**Strengths Contributions:**

+ important problem with low coverage of datasets and benchmarks
+ sound methodology

---

> ### Author Rebuttal · Authors · 2025-07-31
>
> We sincerely thank the reviewer for taking the time to review our paper and for providing valuable comments. Please see our individual responses below:
>
> > difficulty in assessing the precision of the evaluation methodology. The authors could report more details on how precise is the LLM as a judge in evaluating the leakage.
>
> **Response**: We designed the evaluation pipeline so that the judge essentially functions as a fuzzy string matcher. So, we don’t expect that the judge understands what privacy leakage is. For example, let’s say we have the following list of irrelevant sensitive_data: ["Zen Buddhism", "Attended a Temple", "Left Christianity"]. The judge's role is to detect the presence of these “keywords” in the actions produced by the agent. Of course, one could do an exact string match, which is less accurate in this case since the LLM agent may output: “he accepted a branch of Buddhism called Zen”. Moreover, occasionally keywords are high-level concepts (e.g. “changed his religion”). Overall, there is a reasonable expectation that this type of judge is pretty robust.
>
> To further validate this, we conducted a human evaluation for this rebuttal. We selected a single run (gpt-4o with axtree scaffolding) and tasked four human evaluators with assessing whether privacy leakage occurred. We then compared their assessments with the output from the LLM judge. The results showed a 98% agreement between the human evaluators and the LLM judge.

---

> > ### Comment · Reviewer_pff7 · 2025-08-01
> >
> > Thanks for the response.

---

### Official Review · Reviewer_u2GK · 2025-06-28

**Rating:** 5
**Confidence:** 3

**Summary:**

This paper introduces AgentDAM (Agent Data Minimization), a benchmark to evaluate whether AI web agents follow the privacy principle of "data minimization". The benchmark tests if agents inadvertently leak irrelevant private information while executing web-based tasks.

* Built on WebArena/VisualWebArena simulators with 246 realistic web navigation tasks
* Evaluation Approach; Utility: Binary score for task completion + Privacy: LLM-based judge (GPT-4) analyzes agent actions
Main Findings
* Simply asking LLMs about privacy appropriateness overestimates privacy performance by 2-3x
* Agent Performance: Without mitigation: GPT models leak sensitive data 54-75% of the time
*  Leakage Patterns:Most leakages occur in long-text outputs

**Additional Feedback:**

Ultimately the thing that I am reflecting on that gives me pause is how comprehensive this dataset is. Does doing well on this benchmark mean the model is behaving privately? At the moment, I don't have a good sense of failure modes of the benchmark and what it does *not* say (e.g. doing well on the dataset does not mean the model won't do Y). Commenting on this would make me feel much more positively about this work.

**Dataset Code Accessibility:**

Partly

**Dataset Code Comments:**

* The github itself is fairly large. Nearly all of it comes from storing 180MB `ai-agent-privacy/visualwebarena/environment_docker/webarena-homepage/static/input_images` This does not seem necessary, and I believe you can host it somewhere else, that you pull as needed (e.g.hugging face). This also will make it easier to add/remove from the dataset as time emerges

* It took me ~4 attempts to run 	`git clone` for this dataset, I’m not entirely sure why. I suspect this has to do with size of the dataset (I understand this may be a github issue, but I’ve not had this issue with other repos).

* Using the dataset, it would be good to get a sense of how much money something would cost, before just giving a benchmark my API key to run free with it.

* I would add more documentation about how the infrastructure works, using claude code to figure it out helped me a lot, and I would considering adding something like this to your README, I found it extremely clarifying:
```
 What it does:
  AgentDAM evaluates whether AI agents inadvertently leak private information while performing web tasks. It tests agents on realistic web environments (GitLab, Reddit, Shopping sites) to measure privacy
  violations.
  How it works:
  1. Test Setup
    - Creates mock web environments for GitLab, Reddit, and shopping sites
    - Each test case includes:
        - A task intent (what the agent should do)
      - Sensitive data that should NOT be exposed
      - Expected completion criteria
  2. Agent Execution
    - The AI agent (GPT-4o, Llama, etc.) navigates the web environment
    - It receives observations (accessibility tree or screenshots)
    - Takes actions: click, type, scroll, etc.
    - Each action is converted to a text description
  3. Privacy Monitoring
    - A separate "Privacy Judge" (also GPT-4o) evaluates EVERY action
    - It checks if the action reveals any sensitive data
    - Scores: 0 = no leak, 1 = privacy violation
    - Example: If sensitive data includes "Alice Doe" and the agent types "@alice_doe", that's a violation
  4. Evaluation Process
    - Runs through multiple test scenarios
    - Tracks both task completion AND privacy violations
    - Generates reports showing:
        - Task success rate
      - Number of privacy leaks
      - Detailed trajectories (saved as HTML)
  5. Privacy Detection Logic
    - Uses semantic matching (not just exact string matching)
    - Example: "medications against itching" → "antihistamine" = violation
    - Ignores anonymized references like "Customer #1"
```

* Generally pretty thorough set up.
* One question that would be interesting to explore is how the different privacy categories interact with one another? does making something more "private" help all categories unilaterally? Are there some form of trade off (e.g. increasing medical privacy is harder than educational, or intrudes on another form of privacy, if you are maintaining utility).

**Ethical Considerations:**

No, there are no or only very minor ethics concerns

**Final Justification:**

I think that this is good work, and does a good job of addressing in an unaddressed need. As mentioned in my comment, my primary meta-advice here is that taking steps in increasing usability will make it an increasingly meaningful contribution to the literature; the easier it is to pick-up-and-use the more people will pick-it-up-and-use-it.
As my score is already high, I leave it is as is.

**Limitations Weaknesses:**

* You use GPT-4o as the privacy judge to evaluate other models (including GPT-4o itself), this creates a potential bias where the judge's understanding of privacy may align with its own behavior. There is no validation that the LLM judge accurately captures human privacy expectations

* Synthetic Data Limitations - All sensitive data is artificially constructed and inserted into scenarios, would be good to translate these results for at least some real world effects, or at least explain why this is isn’t necessary

* No gradation of sensitivity - treats all categories equally. I likely want more analysis of the different kinds of privacies, and surfacing that.

* Question - Do you feel that 3 runs per experiment is enough? Is there a reason to not do more?

* Please define "probing" better - I understand you aren't introducing the concept, but there is a term in interpretability for "probing" a model, that involves looking the embeddings at some specific layer. It is similar enough in origin that I think you should explain/define what you mean by probing to avoid confusion (especiaily since you even use the term in the abstract where people might not be clued into the context as well.)

**Strengths Contributions:**

Motivation:
* Clear Problem Identification - I really like the problem and think it's great.

Novel Contribution

* First End-to-End Privacy Benchmark for Web Agents
* Important Insights about the specific data minimization practices of these models

Technical Strengths
* Realistic Evaluation Environment
* Automated Evaluation Pipeline
* Multi-Model Evaluation
* Systematic Mitigation Analysis

---

> ### Author Rebuttal · Authors · 2025-07-31
>
> We sincerely thank the reviewer for taking the time to review our paper and for providing valuable comments. Please see our individual responses below:
>
> > You use GPT-4o as the privacy judge to evaluate other models (including GPT-4o itself), this creates a potential bias where the judge's understanding of privacy may align with its own behavior. There is no validation that the LLM judge accurately captures human privacy expectations
>
> **Response**: We designed the evaluation pipeline so that the judge essentially functions as a fuzzy string matcher. So, we don’t expect that the judge understands what privacy leakage is. For example, let’s say we have the following list of irrelevant sensitive_data: ["Zen Buddhism", "Attended a Temple", "Left Christianity"]. The judge's role is to detect the presence of these “keywords” in the actions produced by the agent. Of course, one could do an exact string match, which is less accurate in this case since the LLM agent may output: “he accepted a branch of Buddhism called Zen”. Moreover, occasionally keywords can be high-level concepts (e.g. “changed his religion”). This setup ensures no interaction between models, and we're confident that this approach minimizes any potential for bias.
>
> To further validate this, we conducted a human evaluation for this rebuttal. We selected a single run (gpt-4o with axtree scaffolding) and tasked four human evaluators with assessing whether privacy leakage occurred. We then compared their assessments with the output from the LLM judge. The results showed a 98% agreement between the human evaluators and the LLM judge.
>
> > Synthetic Data Limitations - All sensitive data is artificially constructed and inserted into scenarios, would be good to translate these results for at least some real world effects, or at least explain why this is isn’t necessary
>
> **Response**: We intentionally designed the benchmark using synthetic sensitive data to ensure that the dataset could be safely released for community use without raising ethical or privacy concerns. Even if certain sensitive information is publicly available on the web, including real user data in a benchmark poses significant ethical risks and could inadvertently expose individuals. To strike a balance between realism and responsible data sharing, we had annotators create realistic scenario seeds, which were then used to generate synthetic data that closely mimics real-world contexts. This approach allows us to evaluate privacy risks in a controlled, reproducible manner while enabling open access and broad adoption by the research community.
>
>
> > No gradation of sensitivity - treats all categories equally. I likely want more analysis of the different kinds of privacies, and surfacing that.
>
> **Response**: While it treats all categories of sensitive information equally, our goal was to establish a baseline for privacy evaluation across diverse webnav tasks. This uniform treatment simplifies the initial evaluation process, making it adaptable to various agents and tasks. Our benchmarks's primary goal is to highlight the prevalence of privacy leakage and the need for data minimization, serving as a catalyst for further research. Future work can certainly build on this foundation by introducing sensitivity gradation and more nuanced analyses.
>
>
> > Question - Do you feel that 3 runs per experiment is enough? Is there a reason to not do more?
>
> **Response**: Thanks for bringing this concern. The main reason is that running these agentic models is highly resource-intensive. For example, executing Claude-CUA on our entire benchmark takes approximately 30 hours (without parallelism).
>
>
> > Please define "probing" better - I understand you aren't introducing the concept, but there is a term in interpretability for "probing" a model, that involves looking the embeddings at some specific layer. It is similar enough in origin that I think you should explain/define what you mean by probing to avoid confusion (especiaily since you even use the term in the abstract where people might not be clued into the context as well.)
>
> **Response**: Thanks for sharing your concerns! Here, by probing we mean prompting LLM with user_task+user_data and asking if it is okay to share sensitive_data to finish the task. We will clarify this in the next revision.
>
>
> > The github itself is fairly large. Nearly all of it comes from storing 180MB…It took me ~4 attempts to run git clone for this dataset, I’m not entirely sure why. I suspect this has to do with size of the dataset (I understand this may be a github issue, but I’ve not had this issue with other repos).
>
> **Response**: Thank you for taking the time to try out our codebase. We believe the main issue might be the reliance on (visual)webarena benchmark, which is somewhat heavy. We will remove unnecessary dependencies and refactor our repository to simplify things.
>
>
> > Using the dataset, it would be good to get a sense of how much money something would cost, before just giving a benchmark my API key to run free with it.
>
> **Response**: Excellent suggestion! We'll include this discussion. A rough calculation based on the number of input/output tokens shows that running GPT-4o would cost approximately 60 USD per run. In contrast, Claude CUA was somewhat more expensive, with a 30-hour run costing around 1,200 USD. This partially explains why we did not scale up our dataset to thousands of samples. This also highlights the challenges in designing benchmarks for AI agents, which we, as a community, hope to address in the near future.
>
>
> > I would add more documentation about how the infrastructure works, using claude code to figure it out helped me a lot, and I would considering adding something like this to your README, I found it extremely clarifying:......
>
> **Response**: Thank you for this detailed suggestion! We will incorporate your input into README.
>
>
> > One question that would be interesting to explore is how the different privacy categories interact with one another? does making something more "private" help all categories unilaterally? Are there some form of trade off (e.g. increasing medical privacy is harder than educational, or intrudes on another form of privacy, if you are maintaining utility).
>
> **Response**: We analyzed privacy leakages across different task types (section 4.4) and found out that longer-text outputs are more prone to privacy leakage. It appears that certain task types, such as free-form filling, inherently generate longer outputs, which in turn lead to more leakages. This is reasonable, as the nature of these tasks involves more extensive text. Expanding this to different privacy categories would be very interesting. For example, our initial observations suggest that the agents are indeed more hesitant to share sensitive categories like medical data compared to (subjectively) less sensitive ones, such as information about religious celebrations. However, we do not have precise numbers to confirm this.

---

> > ### Comment · Reviewer_u2GK · 2025-08-03
> >
> > Great - thank you for your responses.
> > I think that this is good work, my primary meta-advice here is that taking steps in increasing usability will make it an increasingly meaningful contribution to the literature; the easier it is to pick-up-and-use the more people will pick-it-up-and-use-it.
> >
> > As my score is already high, I leave it is as is.

---

### Official Review · Reviewer_znFg · 2025-07-01

**Rating:** 5
**Confidence:** 4

**Summary:**

LLM-based AI agents can have access to sensitive user data that they may unnecessarily leak. Therefore, benchmarking the privacy leakage of LLM agents is essential. AgentDAM is a privacy-benchmark for web agents which contains a dataset of human-generated user instructions, synthetically generated user data, and ground truth of sensitive data that is irrelevant to the task. The authors use the benchmark to evaluate web agents based on commonly used LLMs on Visual WebArena tasks demonstrating the privacy-leakage in state-of-the-art LLMs.

**Additional Feedback:**

Thank you for submitting to NeurIPS D&B. I believe this is a nice contribution to the field and I will be happy to increase the score if the authors address the weaknesses mentioned above.

I have one question for the authors. Since the numbers in Table 5 are integers, are these results from a single run? Wouldn't the variance across runs be huge?

**Dataset Code Accessibility:**

Yes

**Dataset Code Comments:**

The code and dataset are available on Github.

**Ethical Considerations:**

No, there are no or only very minor ethics concerns

**Final Justification:**

I believe all reviewers are more or less in sync on this paper. The work has a good technical contribution. I believe the dataset will definitely be used by future researchers. There was some missing information in the original submission, for example, an explicit threat model. The authors have promised to update the paper to add these details in the final submission. I have updated my score accordingly. Overall, this is a strong paper.

**Limitations Weaknesses:**

[W1] The benchmark is very close to PrivacyLens and the only difference appears to be the multi-modal nature. This reduces the novelty of the benchmark.

[W2] The threat model assumed in the paper is implicit. The user prompt is public and the LLMs are the cause of leak of privacy. The sensitive attributes should never be leaked. An explicit section with the threat model will make the paper stronger.

[W3] Lines 184-190 put forward a crucial point that the SENSITIVE_DATA must be irrelevant. I believe this is one strength of the paper. However, it is not discussed in the paper how it was achieved and how successful the approach was. It is an important point that should be discussed in detail.

[W4] Privacy measurement (evaluation) is done using LLM-as-a-judge. This is a probabilistic approach that should be correct in most scenarios but cannot be fully trusted.

[W5] Section 4.2 and the caption of Table 3 states that the these are privacy scores presented in the table. However, the AgentDAM column is taken from the utility column and not the privacy column of Table 4. I believe this is a typo.

[W6] In the experiments, the variance across the three runs is not presented.

**Strengths Contributions:**

[S1] Well-written and easy-to-follow paper.

[S2] The addressed problem of privacy in AI agents is very timely and important.

[S3] The benchmark supports multi-modal web browsing tasks.

[S4] I like the fact that The SENSITIVE_DATA attribute in the benchmark is completely irrelevant to the task, making evaluations easy to interpret.

---

> ### Author Rebuttal · Authors · 2025-07-31
>
> We sincerely thank the reviewer for taking the time to review our paper and for providing valuable comments. Please see our individual responses below:
>
> > [W1] The benchmark is very close to PrivacyLens and the only difference appears to be the multi-modal nature. This reduces the novelty of the benchmark.
>
> **Response**: While AgentDAM and PrivacyLens both evaluate privacy leakage in agentic settings, our benchmark introduces several key differences that go beyond simply being multi-modal. First, AgentDAM operates in realistic, fully functional web environments rather than emulated, text-only settings, enabling end-to-end evaluation of agents' actions in a controllable yet realistic context. Second, it supports both multi-modal (text + image) and text-only inputs. Third, the benchmark covers a diverse set of agentic tasks across multiple real-world web applications (Reddit, GitLab, shopping), with carefully constructed scenarios that embed both relevant and irrelevant sensitive information, enabling nuanced measurement of data minimization in action. Finally, our benchmark proposes and systematically evaluates privacy mitigation strategies, and provides detailed analysis of leakage patterns and mitigation effectiveness. These aspects collectively make AgentDAM a more comprehensive and realistic benchmark for privacy evaluation in autonomous web agents.
>
>
> > [W2] The threat model assumed in the paper is implicit. The user prompt is public and the LLMs are the cause of leak of privacy. The sensitive attributes should never be leaked. An explicit section with the threat model will make the paper stronger.
>
> **Response**: In section 2, we highlight that we consider benign settings without adversaries. We will make it more explicit by adding a separate section/paragraph describing the threat model.
>
>
> > [W3] Lines 184-190 put forward a crucial point that the SENSITIVE_DATA must be irrelevant. I believe this is one strength of the paper. However, it is not discussed in the paper how it was achieved and how successful the approach was. It is an important point that should be discussed in detail.
>
> **Response**: In Figure 3, we illustrate our process where human annotators first design Tasks for agents and create a related Plot, like a chat or document, needed to complete the task. Next, they need to come up with a set of unrelated sensitive_data, such as ["Zen Buddhism", "Attended a Temple", "Left Christianity"] and incorporate them into the Plot while making sure it is irrelevant to the Task. We then use a LM to generate a dataset based on the Task+Plot. Finally, we manually review the data to ensure there are no unclear cases.
>
>
> > [W4] Privacy measurement (evaluation) is done using LLM-as-a-judge. This is a probabilistic approach that should be correct in most scenarios but cannot be fully trusted.
>
> **Response**: We designed the evaluation pipeline so that the judge essentially functions as a fuzzy string matcher. So, we don’t expect that the judge understands what privacy leakage is. For example, let’s say we have the following list of irrelevant sensitive_data: ["Zen Buddhism", "Attended a Temple", "Left Christianity"]. The judge's role is to detect the presence of these “keywords” in the actions produced by the agent. Of course, one could do an exact string match, which is less accurate in this case since the LLM agent may output: “he accepted a branch of Buddhism called Zen”. Moreover, occasionally keywords can be high-level concepts (e.g. “changed his religion”). This setup ensures no interaction between models, and we're confident that this approach minimizes any potential for bias.
>
> To further validate this, we conducted a human evaluation for this rebuttal. We selected a single run (gpt-4o with axtree scaffolding) and tasked four human evaluators with assessing whether privacy leakage occurred. We then compared their assessments with the output from the LLM judge. The results showed a 98% agreement between the human evaluators and the LLM judge.
>
>
> > [W5] Section 4.2 and the caption of Table 3 states that the these are privacy scores presented in the table. However, the AgentDAM column is taken from the utility column and not the privacy column of Table 4. I believe this is a typo.
>
> **Response**: Correct, we made a mistake when copying the results and will fix it. This doesn’t change our overall conclusion, although privacy scores on Llama models seem to be similar. We will adjust the text accordingly. Thanks for pointing this out!
>
>
> > [W6] In the experiments, the variance across the three runs is not presented.
>
> **Response**: The variances are between 2-5% across all models so we omitted it. We will add them to the Appendix.
>
>
> > I have one question for the authors. Since the numbers in Table 5 are integers, are these results from a single run? Wouldn't the variance across runs be huge?
>
> **Response**: Yes, we ran the experiment once and did not observe a significant variance across runs. We tried to structure the tasks in a way that minimizes randomness and variability, leading to more consistent results across different runs.

---

> > ### Comment · Reviewer_znFg · 2025-08-03
> >
> > Thank you for participating in the rebuttal and for responding to my comments.
> >
> > I just have a few comments:
> >
> > 1. In [W2], the threat model does not mean only the presence or absence of adversaries. Please also explicitly include what data is sensitive and why that would be leaked (in this case, for example, the stochasticity of the LLMs), and what information is public and can be leaked.
> > 2. Since human annotators can make mistakes, how can the authors be sure that the SENSITIVE_DATA is completely irrelevant to the task?
> > 3. "We tried to structure the tasks in a way that minimizes randomness and variability, leading to more consistent results across different runs." What does this mean, and how was this achieved?

---

> > > ### Author Response · Authors · 2025-08-04
> > >
> > > Thanks for reading our rebuttal and for additional comments.
> > >
> > > > 1. In [W2], the threat model does not mean only the presence or absence of adversaries. Please also explicitly include what data is sensitive and why that would be leaked (in this case, for example, the stochasticity of the LLMs), and what information is public and can be leaked.
> > >
> > > **Re**: You're right, we'll make it explicit in our paper. Currently all pieces are there but might be scattered, e.g. sensitive_data is described in Section 3.2, what can be shared vs what is considered as a leakage are in Sections 3.2-3.3 / Fig. 2.
> > >
> > > > 2. Since human annotators can make mistakes, how can the authors be sure that the SENSITIVE_DATA is completely irrelevant to the task?
> > >
> > > **Re**: Once all user_tasks, user_data and plots are generated, we perform second round of human annotations, to ensure that SENSITIVE_DATA is irrelevant. We also task annotators to exclude ambiguous cases from the dataset. While we recognize that human error cannot be entirely eliminated, we believe this two-stage process significantly reduces the risk.
> > >
> > > > 3. "We tried to structure the tasks in a way that minimizes randomness and variability, leading to more consistent results across different runs." What does this mean, and how was this achieved?
> > >
> > > **Re**: we mean that we have taken deliberate steps to reduce sources of stochasticity that could affect experimental outcomes. Specifically, we:
> > > - we fix the random seeds and set the temperature for LMs to zero.
> > > - carefully design the set of tasks (Table 6). The tasks are more or less straightforward and capable agents are generally follow the same steps to accomplish them.
> > > - there are not changes in the environment across repetitions (i.e. re reset the environment after each run).
> > > - we assume that the sate transition function S in MDP is static. This assumption may fail sometimes, e.g. some CSS elements fail to load, randomness in the playwright browser interaction framework, etc., but these are rare.
> > >
> > > We acknowledge that these steps do not eliminate stochasticity entirely. Our main point is that, in practice, we do not observe significant variance across runs. To quantify this, we conducted most experiments in our paper three times and report the average results. For the specific experiment presented in Table 5, we initially ran it once for the paper submission. However, during the rebuttal phase, we re-ran this experiment and observed that the variance across three runs was between 3% and 6%.

---

> > > > ### Comment · Reviewer_znFg · 2025-08-04
> > > >
> > > > Thank you for the clarifications. I will increase my score based on the rebuttal.

---

### Official Review · Reviewer_MpbW · 2025-07-03

**Rating:** 5
**Confidence:** 3

**Summary:**

This paper introduces AGENTDAM, a novel benchmark designed to evaluate whether autonomous web agents adhere to the privacy principle of "data minimization." Key contributions include:
- **A New Privacy-Centric Benchmark**: AGENTDAM is the early benchmark to specifically measure if web agents use a user's potentially sensitive information only when it is strictly "necessary" to complete a task.
- **Realistic and Reproducible Environment**: The benchmark is built on the existing WEBARENA and VISUALWEBARENA frameworks, enabling end-to-end evaluation of agent behavior in realistic, yet controlled and reproducible, web environments.
- **Comprehensive Task and Evaluation Design**: It consists of 246 tasks across three websites (Reddit, GitLab, Shopping), where each task provides the agent with both relevant and irrelevant sensitive information. Performance is measured along two axes: task success (utility) and privacy leakage, with an LLM-as-a-judge system designed to automate the detection of nuanced privacy violations.
- **Key Findings and Mitigation**: The authors evaluate several state-of-the-art models (GPT-4, Llama-3, Claude) and find they are all prone to leaking unnecessary sensitive information. They also propose and validate a prompting-based mitigation strategy that effectively reduces leakage.

**Dataset Code Accessibility:**

Yes

**Dataset Code Comments:**

The dataset and code are highly accessible. The authors provide the task definitions, data, and evaluation methodology directly on GitHub, which fosters transparency and allows the community to easily reproduce the results and build upon this work.

**Ethical Considerations:**

No, there are no or only very minor ethics concerns

**Limitations Weaknesses:**

The primary weakness of the paper is the lack of a human baseline, which would help to contextualize the results.
- **Absence of Detailed Human Evaluation**: The paper lacks a formal human performance baseline for the tasks. While annotators created the data seeds, it's unclear if a human can consistently complete all 246 tasks with perfect privacy and utility. Establishing a human baseline would be critical for two reasons: 1) It would validate that the tasks are indeed solvable without leaking sensitive information, confirming the soundness of the task design. 2) It would provide a crucial reference point for interpreting the agents' scores, helping the community to better understand the gap between current AI capabilities and desired human-level performance in data minimization.

**Strengths Contributions:**

This work makes several important contributions to the field of AI safety and agent evaluation.
- **Fills a Critical Research Gap**: The paper addresses the crucial but under-explored problem of inference-time privacy for AI agents. By shifting the focus from training data privacy to the agent's active handling of user data, AGENTDAM provides a framework for a new and highly practical area of evaluation.
- **Provides Valuable Insights**: The experiments yield significant findings, such as the clear difference in baseline privacy awareness across different model families and the fact that even the best-performing models still have considerable room for improvement. The discovery that simply probing an LLM about privacy overestimates its actual performance in an agentic setting is a particularly important insight that validates the need for end-to-end benchmarks like this one.
- **Offers and Evaluates a Simple Straightforward yet Practical Defense**: The paper does not stop at identifying the problem but also proposes a straightforward and effective prompting-based mitigation strategy. The detailed evaluation of this method, showing it can significantly improve privacy with a minor trade-off in utility, is a valuable contribution for developers aiming to build more responsible agents.

---

> ### Author Rebuttal · Authors · 2025-07-31
>
> We sincerely thank the reviewer for taking the time to review our paper and for providing valuable comments. Please see our individual responses below:
>
> > Absence of Detailed Human Evaluation: The paper lacks a formal human performance baseline for the tasks....
>
> **Response**: While adding a user study and collecting/releasing human traces are on our roadmap, these activities are not feasible within the rebuttal period. One reason we believe human performance on this task (in terms of privacy) will be high is that we meticulously created SENSITIVE_DATA to be completely irrelevant to the task, as also noted by reviewer znFg. We also performed manual filtering of ambiguous cases. This setup should make it straightforward for humans to recognize and disregard information that is clearly unrelated to the task at hand.

---

> > ### Comment · Reviewer_MpbW · 2025-08-05
> >
> > Thanks for the response.

---

> > > ### Comment · Reviewer_pff7 · 2025-08-05
> > >
> > > I wanted to confirm that I have read the rebuttal and I don't have further questions or comments for the authors.

---

### Comment · Area_Chair_FmUh · 2025-08-08

Dear Authors and Reviewers,

The deadline for author-reviewer discussions is approaching. If you haven’t done so already, please ensure there are sufficient discussions for both the submission and the rebuttal.

Reviewers, please make sure you complete the mandatory acknowledgment AND respond to the authors’ rebuttal, as requested in the email from the program chairs.

Authors, if you feel that any results need to be discussed and clarified, please notify the reviewer. Be concise about the issue you want to discuss.

Your AC

---

### Decision · Program_Chairs · 2025-09-18

**Decision:**

Accept (poster)

**Comment:**

All reviewers have provided positive scores for this submission, highlighting its strengths and key contributions. Given the unanimous positive feedback and the recognition of its contribution to the area, the AC carefully reviewed the paper and concurred with the reviewers' assessments, therefore supporting the decision to accept this submission.